# Young adults' perceptions of living with atopic dermatitis in relation to the concept of self-management: a qualitative study

Susanne Lundin ,[1,2] Marina Jonsson,[2,3] Carl-Fredrik Wahlgren,[4] Emma Johansson,[4,5] Anna Bergstrom,[6,7] Inger Kull[1,2]

For numbered affiliations see end of article.

**Correspondence to**
Mrs Susanne Lundin;
susanne.lundin@ki.se

## ABSTRACT

**Objectives** Learning to take control of one's health is an important part of the transition from adolescence to adulthood. This study aimed to explore young adults' perceptions of living with atopic dermatitis (AD) in relation to the concept of self-management.

**Design** A qualitative study with an inductive approach was performed through semistructured interviews (n=15). The interviews were recorded, transcribed verbatim and analysed with systematic text condensation.

**Participants** Young adults (mean age 23,4 years) with persistent AD in a longitudinal population-based birth cohort. To capture experience of living with persistent AD (preschool/school-age onset) of different severity (mild to severe/very severe), a purposive selection was performed. In total, 15 young adults were included. Persistent AD (preschool/school-age onset) was defined as dry skin in combination with itchy rash of typical localisation in the 12 months preceding the 16-year and the 24-year follow-ups. Severity was self-assessed using the Patient Oriented Eczema Measure.

**Results** Despite having experience of AD since childhood, the respondents expressed uncertainty about treatment and how it affected their bodies. Their uncertainties and feelings affected how they used topical corticosteroids. The respondents emphasised that they perceived availability of healthcare and knowledge about treatment of AD among healthcare providers to be limited. The participants did not state any experiences of support to self-management from healthcare, which affect young adults' possibilities to take full control of their AD care.

**Conclusions** Young adults with preschool/school-age onset of AD are unsure how to treat and manage the disease. One explanation may be insufficient transition process.

## INTRODUCTION

To optimise self-management in young people with chronic disease, there is a need for a transition process to increase knowledge and to achieve self-care skills and independence.[1 2] To aid young adults to be autonomous and minimising dependency from parents and healthcare regarding self-management of eczema, it is important that the transition process starts at an early age (11–13 years) in collaboration with the family and all stakeholders.[3] Atopic dermatitis (AD)[4] is a chronic disease that cannot be cured

## Strengths and limitations of this study

► Data from a longitudinal population-based birth cohort, with the possibility to obtaining detailed information about different phenotypes and severity of atopic dermatitis (AD) over time.
► The AD severity ranged from mild to severe/very severe as, measured by Patient Oriented Eczema Measure, a validated questionnaire.
► Equal representation of young men and women.
► The results may not be transferred to young adults with more severe AD requiring specialist care.

as yet and requires daily treatment.[5] In addition, it is known that AD affects health-related quality of life in both children and adults.[6 7] The symptoms range from mild to very severe.[8] Most individuals with AD have a mild disease.[6 9] However, there seems to be a discordance between patients' and physicians' assessments of severity judged using different instruments.[10 11] The patients reported AD as more severe when their quality of life was impaired, while the physicians reported AD as more severe if sleep was reported to be disturbed.[6 7] To ensure self-management in AD, it is important that the individuals have sufficient skills related to treatment principles and are aware of the goals of treatment, which are to increase disease control and to extend asymptomatic periods as much as possible.[6 12] This requires knowledge and education. The mainstay of treatment is the use of emollients everyday to decrease dryness and extend time to relapse of AD.[8] The first-line treatment of inflammation consists of topical corticosteroids (TCSs) with differing potency depending on age, localisation and severity of the symptoms.[12] The chronic, recurrent nature of the disease,[6] and the often time-consuming treatment, to be performed everyday, can be a challenge for individuals with AD.[13] Since parents have the main responsibility for the treatment of AD in their children, it is of importance that the parents have been trained to and understand the best

management of the disease.[8] A shift in responsibility takes place when the children reach adolescence, which requires that the adolescents gradually develop knowledge of their treatment. However, results from a Swedish population-based birth cohort show that many adolescents with AD are untreated.[14]

In Sweden, special transition models have been developed for chronic diseases such as congenital heart diseases and diabetes (still under evaluation)[15 16] but—to the best of our knowledge—not for young persons with AD. According to the recently published guidelines on effective transition by the European Academy of Allergy and Clinical Immunology,[3] learning to take control of treatment is an important part of the transition from adolescence to adulthood. Overall experiences of how young adults (from the general population) with AD manage the disease are scarcely studied. The aim of this qualitative study was, therefore, to explore young adults' perceptions of living with AD in relation to the concept of self-management.

## METHODS
### Design
A qualitative semistructured interview study with an inductive approach was used to capture young adults' experiences of AD in the concept of self-management. We followed the Consolidated Criteria for Reporting Qualitative Research guideline for conducting and reporting qualitative research.[17]

### Setting
The present study is a part of the longitudinal population-based cohort BAMSE (Children, Allergy, Environmental, Stockholm and Epidemiology).[18 19] A purposive selection was performed to capture young adults with experience of persistent AD (preschool/school-age onset), with various severities (mild to severe/very severe) (table 1).

Moreover, we strived to capture an equal proportion of men and women with different severity of AD. They were invited to the interview study by a clinical research nurse over the telephone. Thereafter, the first author contacted the participants, informed them about the study and asked whether they wanted to share their experiences of living with AD.

### Inclusion criteria
Young adults with persistent AD, defined as dry skin in combination with itchy rash of typical localisation (neck/throat, arm/leg flexures and/or wrists/ankles) in the 12 months preceding the 16-year and the 24-year follow-ups. The severity of AD was self-assessed as mild to severe/very severe using the Patient Oriented Eczema Measure,[20] which enabled us to capture a range of experiences. The young adults who fulfilled both the questionnaire definition of AD (as above) and the clinical definition of AD; visible AD on the neck/throat and/or flexure surfaces of the arms/legs and/or wrists/ankles and/or reported symptoms of itchy skin in the preceding week as assessed by a trained nurse were included.

| Table 1 | Clinical description of sex, severity and onset of AD among the participants | | | |
|---|---|---|---|---|
| Sex | Self-assessed severity at 16 years * | Self-assessed severity at 24 years * | Onset of eczema up to 4 years † | Onset of eczema from 8 years ‡ |
| M | Mild | Moderate | | X |
| F | Mild | Moderate | X | |
| M | Moderate | Moderate | X | |
| M | Mild | Moderate | X | |
| M | Mild | Mild | | X |
| F | Mild | Mild | | X |
| M | Moderate | Severe/very severe | X | |
| M | Moderate | Mild | X | |
| F | Mild | Moderate | X | |
| F | Mild | Severe/very severe | X | |
| F | Severe/very severe | Moderate | X | |
| M | Mild | Severe/very severe | X | |
| F | Moderate | Moderate | X | |
| F | Mild | Severe/very severe | | X |
| M | Severe/very severe | Severe/very severe | X | |

*Patient Oriented Eczema Measure class 1, mild eczema (3–7), class 2, moderate eczema (8–16), class 3, severe/very severe eczema (17–28).
†Preschool eczema (ie, eczema onset at 1,2 or 4, not 8, 12 or 16 years of age).
‡School age eczema (ie, eczema onset at 8, 12 or 16 not at 1,2 or 4 years of age).
AD, atopic dermatitis.

**Table 2** Interview guide

| Themes related to experiences of eczema, treatment and care | Follow-up questions |
|---|---|
| Tell me what it is like to live with eczema. | Tell me what it has been like lately. What was it like when you were a teenager? Is there any difference? |
| Tell me what it is like when you have trouble with your eczema. | What is it like when your eczema is not troubling you? How often do you have trouble with your eczema? What do you think is the cause of your eczema? |
| Tell me how you usually self-treat. | How do you feel it works? How has treatment been shown or explained to you? What do you think about cortisone? What do you think about moisturising cream? |
| Tell me about the experiences you have of healthcare in relation to your eczema. | Tell me about the needs you have from healthcare due to your eczema. Tell me about any information you have received from healthcare regarding your eczema. Could it have been given in any other way? |
| In what situations do you think about your eczema? | Do you avoid doing anything because of your eczema? Have you avoided anything in the past because of your eczema? |

## Data collection

All interviews were conducted at the research clinic, except two, which were conducted by telephone. The interviews were performed by the first author, an experienced dermatology nurse (SL) with no connection to the study participants before the interviews. The semistructured interview guide consisted of five themes related to living with AD: daily life, symptoms, treatment, care and consequences of the disease (table 2).

Probing questions were used to get a deeper understanding or explanation of some statements. To test the interview guide, one pilot interview was conducted (not included). Minor changes were then made to the interview guide. During the interview, the participants were asked if their answers were interpreted correctly. Thus, the interviewer received confirmation from the respondents. The interviews lasted on average 31 (range 23–43) min. Data collection continued until information power was judged to have been achieved. This is an approach that considers the aim, the sample specificity, the theoretical background, the quality of dialogue and the strategy for analyses. Taken together, these determine whether sufficient information power has been obtained. In total, 15 young adults, mean age 23,4 years were included (table 1), which gave us rich data covering the purpose of the study. All interviews were recorded and transcribed verbatim.

## Data analysis

The text was analysed by three researchers (SL, MJ and IK) in collaboration, using systematic text condensation (STC), a thematic cross-case strategy in four steps.[21] First, all interviews were read through (SL, MJ and IK), and preliminary themes were identified based on the contents of the text (SL and MJ). In the second step, meanings units were identified based on the contents of the preliminary themes and code groups were developed (SL and MJ). In the third step, the material from each code group was decontextualised, sorted into subgroups and the contents of the text were condensed (SL, MJ and IK). Finally, in the fourth step, data were reconceptualised, and categories made based on the essence of the young adults' experiences of eczema (SL, MJ and IK) (table 3).

The analysis was inductive, without any theory-driven approach to the phenomenon of self-management. We were open to the data and what the material told us, from an individual to a general level. In the analysis process, we tried to bracket our preconceptions to minimise the impact of our professional and clinical experiences. Using STC, with many similarities to the editing analysis style,[22] we developed categories derived from the empirical data.

## Patient and public involvement

The primary intention of this study was to capture young adults' perceptions of living with AD in relation to self-management. However, the young adults were not involved in the design of the study.

## RESULTS

The respondents' perceptions of living with AD in the concept of self-management resulted in four categories and eight subgroups (table 3). Despite having had AD since childhood, the respondents expressed uncertainty about treatment and how it could affect their bodies. Moreover, both symptoms and treatment were described as painful. Their uncertainties and feelings affected how they used TCSs. Though the disease made itself known all the time, most of the participants stated that they had gotten used to living with eczema, it was a part of their life. The respondents emphasised that they experienced that both the accessibility and the support from healthcare providers were limited.

**Table 3** The process of analysing young adults' experiences of living with AD with systematic text condensation

| Step 1. preliminary themes | Step 2. codes | Step 3. subgroups | Step 4. categories |
|---|---|---|---|
| *Reminds you* | *Always on my mind*<br>*Almost never symptom-free*<br>*As long as it is not visible* | *Strategies*<br>*Acceptance* | *A part of life* |
| *The care is vague* | *They do not have time*<br>*Everyone says different things*<br>*Lack of information*<br>*The treatment does not work* | *Limited availability*<br>*Limited support*<br>*Uncertainty*<br>*Need for information* | *Difficult to get help*<br>*Limited knowledge* |
| *Physical effects* | *Not good for my body*<br>*Being careful*<br>*Burning and stinging* | *Unnatural*<br>*It hurts* | *Impact on my body* |

### Limited knowledge

Although most of the respondents had had AD from an early age, they expressed that their knowledge about its cause and treatment was limited. They had created their own treatment routines, based on their experiences and/or what they have learnt from their parents. The young adults stated that they experienced a lack of information and directives on treatment.

### Uncertainty

The young adults expressed uncertainty about the cause of their AD, how they should treat it, what they should use to treat it, where on the body and for how long. They had not received much information about the cause or variability of the disease or about different treatments.

> The doctors have been more like it will go away when you reach a certain age or something, it's just now, it will disappear, so… I'm just waiting for that age to come (9)

They felt uncertain about the treatment, if they were doing the right thing, and if the topical treatment should include or consist of cortisone.

> They say cortisone is what helps, I don't know, there's no cortisone in Mometasone, as far as I know (3)

### Need for information

There was a general belief among respondents that TCSs attenuated or diminished symptoms only for the moment, which reinforced the belief that they could have 'become immune' to these creams.

Most of the young adults said they treated themselves regularly, but it was clear that they were primarily referring to moisturising creams. The respondents mentioned different treatment recommendations they had been given over the years, such as not to use TCSs around the eyes and not to apply cream if it was cold outside. They also reported that they were advised to use TCSs sparingly, at most two times a week, without regards to the severity of the AD.

The respondents expressed that it would be helpful with more information and requested a guide with treatment principles, depending on the severity of the AD.

This would be important in order to achieve independence regarding self-management of the disease.

> I have understood by, like, reading and talking with different people that you should have different, like, cycles for it, but I haven't really understood how to do it, because I have not got that input from a doctor, how I should do it (14)

### Impact on my body

Both symptoms and treatment of the AD had a painful impact on the body. Moreover, the need to use different creams to treat the AD was something that felt unnatural and there were thoughts on various treatments adversely affecting the body.

### It hurts

Many of the respondents stated that one of the most difficult things about AD was that it hurts. They described the pain as a burning and stinging feeling under the skin. That pain was annoying and could prevent them from doing things that they liked to do. The pain was sometimes more trying than the itching.

> Periodically, it hurt too much to put on socks and shoes… (1)

They stated that it was painful to apply cream when the AD was severe; they self-treated to get rid of the pain, but that hurt even more.

> 'It stings, of course, every time I apply the cream' (15)

### Unnatural

Some of the respondents felt that it was unnatural to cover their skin with something that could affect the body negatively, especially in the case of TCSs.

> That you absorb, like, products that the body is not really made to deal with (2)

They had heard that the skin could become damaged and thinner from using TCSs, which made them anxious; they, therefore, used these as little as possible. They felt

that it was strange to use a treatment that could simultaneously damage the skin even if it decreased the AD.

> I try not to use too much, plain and simple, and hope it is…that I'm like doing what you're supposed to (7)

### Difficult to get help

There was a perception among the young adults that it was difficult to get help from healthcare, both making an appointment and getting the 'right' treatment. The respondents did not state any experiences of education in self-management by healthcare. Moreover, they perceived that healthcare providers did not recognise the symptoms, which resulted in insufficient and delayed treatment.

#### Limited availability

Even though most of the respondents had had AD all their life, they stated that they had limited contact with healthcare. It seemed to be difficult to get in touch with healthcare and get appropriate help, even when the AD was severe. Sometimes, it took several months to get an appointment and they needed to spend a lot of time trying to get in touch with the care provider. Many of the respondents could not remember the last time they had had an appointment.

> It was probably in August that I contacted my dermatology department, because it [i.e., eczema] was so very bad…and then I got an appointment for the middle of November (14)

#### Limited support

One respondent had experienced that the symptoms of the AD had not been taken seriously; the physician downplayed the respondent's symptoms and compared them with the more widespread symptoms of others. The participants perceived that physicians in general had limited knowledge about treatment of AD and the only advice they had received was to apply topical creams, without further instructions.

> It seems that the GP doesn't know all that much about eczema…and just prescribes ointments that do not help (8)

Information about how they should treat the eczema also differed from physician to physician.

> Some of them say 'apply it when you need to,' others say, 'apply it like several times a day,' it's like people are guessing a little, they don't have any idea, really (10)

### A part of life

The respondents stated that they had become accustomed to living with AD and had accepted their disease. Even so, they had many strategies about how they should avoid worsening of the AD, which required a lot of routines in the form of regular treatment, avoiding scratching and creating strategies in daily life.

#### Strategies

Most of the respondents thought of and were aware of their AD constantly: when it was itching, when it was visible, when they had to apply cream and so on. They had developed different strategies to avoid worsening of the AD. The respondents perceived it as trying to experience itchiness all the time and they were aware of the importance of not scratching, even though it was difficult. Trying not to scratch required constant vigilance. Scratching was the only thing they wanted to do, but they knew it would immediately result in them being 'punished', through worsening of the AD.

> That's the main thing in how I handle it, that just sitting there thinking that okay, don't scratch, don't scratch, don't scratch, don't scratch, don't scratch, it has become like a habit to try not to do it (15)

The major problem was that the AD never disappeared. They had to make choices in their daily life based on their skin condition. Most of the respondents never felt free of symptoms, which they found trying.

> It's probably all the time I'd say, on different parts, it's like it moves around a bit, where it's, like, worst (10)

The treatment of AD was demanding, according to the respondents. Treatment was a bit awkward and nothing they enjoyed. They had to have a regular routine and a strategy; even then, it took a long time to get rid of the AD and the respondents stated that the creams only helped for the moment.

> But the eczema keeps, like, coming back anyway, even though I have this routine (9)

They always had to carry creams with them, to school, work or when they travelled, and if they forgot, they were 'punished' at once, through worsening of the AD.

> Always having creams with you and applying them is not all that much fun either (3)

#### Acceptance

The respondents emphasised that they had gotten used to living with AD and had accepted their situation, even though it was hard. They did not allow AD to keep them from doing things. It was a part of their life.

> It's, like, something you have, you have to learn to live with it, maybe (7)

While they had accepted their disease, they tried to avoid revealing their AD, in so far as possible, when it was in visible places on their bodies.

> I do not like it to be visible, so usually when I have these red, I usually use long sleeves, to like cover up (13)

## DISCUSSION

This qualitative interview study of young adults with persistent AD from the general population demonstrated several shortcomings in the concept of self-management. Even though the young adults had experience of persistent AD, they were unsure about how to treat the AD. The respondents expressed limited knowledge about the differences between emollients and TCSs and regarding the potency of the latter. Their uncertainty may be related to their perception of difficulties in getting help from healthcare in the terms of limited knowledge and lack of support about treatment of AD among healthcare providers. While acceptance of the disease was high, the respondents never felt completely symptom free. Altogether, this indicates an insufficiency regarding self-management, including knowledge and skills in treatment principles for the disease, which may affect young adults' possibilities to take full control of their AD care.

### Knowledge about treatment of eczema

The young adults' uncertainty regarding how to treat and manage the AD was in accordance with other studies. The uncertainty seems to be found also in self-treating adults, even if their AD is more severe; this can delay the start of treatment in flare-ups.[23] In contrast, adolescents (12–18 years) in a qualitative study[24] expressed satisfaction with the effect of their treatments, even though half of them used TCSs everyday because of low efficiency. Moreover, they had limited knowledge and incorrect beliefs about how the treatment worked. The same beliefs were seen in the present study, but the participants in our study expressed less satisfaction and more often believed that treatment, even with emollients, could harm their body.

### Factors of importance for self-management

The limited contact with and difficulties in getting help from healthcare can to some extent be due to limited knowledge regarding whom to contact. Healthcare in Sweden is divided into primary care, specialist care and highly specialised care.[25] Primary care is responsible for treatment and care of the most common skin diseases, such as AD. Individuals with AD can buy up to 30 g of mild TCSs over the counter from a pharmacy; for larger amounts or more potent TGCs, a prescription is needed. The respondents' opinions that physicians had limited knowledge about treatment is in accordance with the results from a qualitative study including GPs. The GPs stated that they used a trial-and-error approach when prescribing emollients, due to the large assortment of different creams, and that they were reluctant to prescribe strong TCSs to children.[26] The prescribed amounts of emollients in primary care have reportedly been unexpectedly low when compared with guidelines, which was also seen for prescription of TCSs in specialist care.[27 28] To have the possibility to treat AD adequately, the individuals need to have access to appropriate potency and amount of TCSs and to treat the AD daily until the symptoms

clear. Thereafter, the TCS therapy should be changed into proactive treatment two times a week.[8]

Concerns about side effects of TCSs are common. Studies including both children and adults with moderate to severe AD have found that approximately 50% of guardians/patients are reluctant to use TCSs.[29 30] In addition, misleading information from relatives/friends, the internet or pharmacies can lead to poor adherence.[31 32]

Moreover, it has been shown that concerns and uncertainties about side effects exist even among healthcare providers.[33] Both GPs and pharmacists inform patients about the risks of TCSs and that they should be applied sparingly and not over long periods of time.[34] This can affect the potency and amount of TCSs prescribed and strengthen the feeling among individuals with AD that a TCS is something to be careful with.[35 36]

While the young adults in the present study had gotten used to living with AD, they stated that they needed more support. Furthermore, there seemed to be a need for repeated information about the cause, course and treatment of the disease. Several studies have stated the importance of shared decision-making, education and a written action plan,[37–40] which could be useful tools to achieve independence. This is important in the process of developing autonomy in self-management and has been confirmed in a study among youths aged 14–24 years,[41] which showed that 41% wanted to be more involved in their care. For adolescents with asthma, it is known that support during the transition process results in better adherence to treatment, but the healthcare transition in AD is less studied.[42] Guidelines on asthma emphasise the importance of reviewing each patient's inhalation technique at every healthcare visit, to decrease the risk of exacerbations.[43] It would be equally important to review how patients with AD perform self-treatment, to achieve as long symptom-free periods as possible.

### Strengths and limitations

One strength of the present study was that all respondents had well-defined AD based on both prospectively recorded questionnaire data and clinical examinations. Moreover, their AD severity ranged from mild to very severe. Additional strengths are the credibility and transferability based on the wide range of data from the respondents, which we have exemplified using the respondents' own words.

The respondents were from the general population, which is a strength since most patients with AD are treated in primary care. However, the results may not be transferred to patients with more severe AD, who may have more contact with specialist care. Still, some of the respondents in the present study had moderate to very severe AD.

In qualitative studies, the number of participants is generally low. However, this can be compensated by a purposive selection, to get a deeper understanding of the respondents' experiences. We recruited participants with long experience of mild to very severe AD, who wanted to

share their experiences. The data collection gave us varied and rich data on perceptions and experiences of living with AD. An interviewer's preconception about subjects can be both positive and negative. Deep experience can be helpful in the interview situation to promote understanding and when asking probing questions, but can also be negative, leading to the assumed understanding of what informants are saying and an assumption that the interviewer and respondent are thinking about the same thing. Preconceptions can also be negative during the analysis phase, and the bracketing of the preconceptions is a way to try to avoid these influences. Even, so this may be an unattainable goal according to Malterud.[44] To minimise these effects, the analysis was carried out in collaboration with two coauthors with little or no experience in the care of patients with AD.

## CONCLUSIONS

This qualitative interview study of young adults with AD from the general population demonstrated several shortcomings in self-management. Despite long experience of AD, the young adults expressed limited knowledge and uncertainty about treatment. Their uncertainty may be related to their perception of difficulties in getting help from healthcare and of limited support, including knowledge about treatment of AD among healthcare providers. Altogether this may indicate shortcomings in the transition process.

### Author affiliations

[1]Sachs' Children and Youth Hospital, Södersjukhuset, Stockholm, Sweden
[2]Department of Clinical Science and Education, Södersjukhuset, Karolinska Institutet, Stockholm, Sweden
[3]Centre for Occupational and Environmental Medicine, Stockholm County Councíl, Stockholm, Sweden
[4]Department of Dermatology and Venereology Unit, Department of Medicine Solna, Karolinska Institutet, Stockholm, Sweden
[5]Department of Dermatology, Karolinska Universitetssjukhuset i Solna, Stockholm, Sweden
[6]Centre for Occupational and Environmental Medicine, Stockholm County Council, Stockholm, Sweden
[7]Institute of Environmental Medicine, Karolinska Institutet, Stockholm, Sweden

**Acknowledgements** We thank all the participants in the BAMSE cohort, and all the staff involved in the study through the years.

**Contributors** SL, MJ and IK contributed to conception and design of the study. The interviews were managed by SL. The analysis with Systematic Text condensation was conducted by SL, MJ and IK in collaboration. Drafting of the manuscript was conducted by SL with support of IK, C-FW, EJ and AB. All authors participated in critical revision of the manuscript, provided important intellectual input and approved the final version.

**Funding** Funding for this study was provided by the Swedish Asthma and Allergy Association, grant/award number 2017–0007.

**Disclaimer** The funder has not been involved in the design and conduct of the study; collection, management, analysis, and interpretation of the data; preparation, review, or approval of the manuscript; and decision to submit the manuscript for publication.

**Competing interests** EJ reports personal lecture fees from Sanofi-Genzyme, LEO Pharma, Novartis, and ACO. EJ and SL has been part of the advisory board at Sanofi-Genzyme.

**Patient and public involvement** Patients and/or the public were not involved in the design, or conduct, or reporting, or dissemination plans of this research.

**Patient consent for publication** Not required.

**Ethics approval** The study was approved by the regional ethical review board at Karolinska Institutet, Stockholm (approval number 2016/1380-31-2, 2017/395-32). All participants gave informed consent, 13 with written consent and two by telephone.

**Provenance and peer review** Not commissioned; externally peer reviewed.

**Data availability statement** No data are available. Audio files and transcribed text has been deposited anonymously at the BAMSE-secretariat, Institute of Environmental Medicine at Karolinska Institutet, Stockholm, Sweden.

**ORCID iD**
Susanne Lundin http://orcid.org/0000-0003-3193-4722

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
