## [Reviewer comments · BMJ Open]

ARTICLE DETAILS

TITLE (PROVISIONAL)	Young adults' perceptions of living with atopic dermatitis in relation to the concept of self-management - a qualitative study
AUTHORS	Lundin, Susanne; Jonsson, M.; Wahlgren, Carl-Fredrik; Johansson, Emma; Bergstrom, Anna; Kull, Inger

VERSION 1 – REVIEW

REVIEWER	Magin, Parker University of Newcastle, Discipline of General Practice
REVIEW RETURNED	06-Oct-2020

GENERAL COMMENTS	Important contexts for the topic of atopic dermatitis (AD) management are that it is a chronic disease that usually commences quite early in childhood and not infrequently continues into adulthood. An additional important feature is that AD involves a great deal of self-management (quite often involving ongoing regular topical treatment and skin care). Not only is this self-management onerous, it is predicated on a good understanding of the nature and natural history of the condition and on the rationale for topical treatment. Having a child with AD is often demanding and time-consuming for parents, who need the requisite understanding of the condition and commitment to its ongoing management. The transition of this responsibility for care from parents to patients as they acquire age-related preparedness is a particularly important, as well as an interesting, period. The current study aims to explicitly address this transition period, but the overall impression is of a fairly superficial exploration of AD in young adults rather than an exploration of transition. The study looks to have been competently performed. One issue is that the authors describe a very quantitative approach to recruitment: '...a strategic selection was performed.every third of these participants was invited to the interview study from a random list....'. A purposive rather than random sampling would be appropriate for a qualitative study such as this. A maximum variation sample rather than random sample would better meet the study aims. The findings are somewhat disappointing. The Results contains a range of findings which relate to current (early adult) experience of AD. These are (as the authors discuss) consistent with previous qualitative research on AD. And the findings are general – the actual transition process is not explored beyond that the 'participants did not state any experiences of transfer from paediatric to adult healthcare.' And deficits in participants' current knowledge etc were attributed (by the authors) to a failure of transition. The transition from childhood to adult AD is far more complex than the transfer of healthcare system, and current young
--

	adult experience of AD will be subject to multiple influences in addition to the transition process. The result is more a study of AD in early adulthood rather than a study of transition in AD. The findings in themselves are interesting, if not novel. Minor issue. • 'Cortisone' is regularly referred to where it should be 'corticosteroids'.
--	---

REVIEWER	Angelhoff, Charlotte Linkopings universitet Campus Norrkoping, Department of Social and Welfare Studies, Division of Nursing Science
REVIEW RETURNED	12-Oct-2020

GENERAL COMMENTS	Thank you for inviting me to review this study. This study is interesting and important. However, some major revision should be considered. Abstract: Under results you use the abbreviation TGC. Please spell out. Introduction Page 6, line 38. In what way can the treatment be challenge for individuals with AD? Please, give some examples I don't think that the rational is clear enough to explain why this study is needed. Methods: Overall, the methods section needs major revision. In qualitative studies it is important to explain to the reader what approach and methodology the authors used. This is unclear throughout the manuscript, which gives the study low impact. It is not enough to state that it is an inductive approach. The systematic text condensation by Malterud is an analysing process based on psychological phenomenology by Giorgi. Page 7. The setting section is not clear. It is confusing how you recruited adolescents to present study, as you have described the setting of another study, BAMSE. You could write that the study is part of a greater study with a reference to BAMSE instead. Line 54, please clarify what a study nurse is Page 8 I am concerned by the analysing process and how it was performed. Line 35. Was the experienced nurse who did the interviews working with the study participants? Was the nurse skilled in interview technique? How come the authors (researchers) did not perform the interviews? The concept "data saturation" (used at page 8, line 45) is a concept in the method Grounded Theory and implies that the interview is transcribed and analysed before the next interview is performed. This is not the procedure for other analysing methods. Please, clarify if you used this analysing method or how you judged that the sample size was enough for your study. Page 8 line 43. Was the pilot interview included in the study? Did you make any changes in the interview guide? Page 9 The four steps of Malteruds analysing process is well described from the reference. However, Figure 1 could include examples of how you did this by including text from your data.
--

	Ethics line 38, remove “ed” in consent: All participants gave informed consent. Results Page 10 line 24 “This indicates insufficiency...” would be better placed in the discussion. You have the subgroup Ignorance and turn it into the category Individual understanding and knowledge. This step is not understandable. Are there more subgroups that are not named? The same with the category Availability and knowledge in health care with the subgroup Difficult to get help. I think the analysing process needs one more turn to set the categories, before the result section is clear. Therefore, I decided not to review the discussion part. Table 2. Background factors... I don't understand why you include so many different measures, what do they add to the study? It would be more interesting to read more demographics, as working status, marital status, educational status... Also, I wonder about the self-assessed severity at 24 years. Some of the participants are 22-23 years old? Thank you for inviting me to review this study. This study is interesting and important. However, some major revision should be considered. Abstract: Under results you use the abbreviation TGC. Please spell out. Introduction Page 6, line 38. In what way can the treatment be challenge for individuals with AD? Please, give some examples I don't think that the rational is clear enough to explain why this study is needed. Methods: Overall, the methods section needs major revision. In qualitative studies it is important to explain to the reader what approach and methodology the authors used. This is unclear throughout the manuscript, which gives the study low impact. It is not enough to state that it is an inductive approach. The systematic text condensation by Malterud is an analysing process based on psychological phenomenology by Giorgi. Page 7. The setting section is not clear. It is confusing how you recruited adolescents to present study, as you have described the setting of another study, BAMSE. You could write that the study is part of a greater study with a reference to BAMSE instead. Line 54, please clarify what a study nurse is Page 8 I am concerned by the analysing process and how it was performed. Line 35. Was the experienced nurse who did the interviews working with the study participants? Was the nurse skilled in interview technique? How come the authors (researchers) did not perform the interviews? The concept “data saturation” (used at page 8, line 45) is a concept in the method Grounded Theory and implies that the interview is transcribed and analysed before the next interview is performed. This is not the procedure for other analysing methods. Please, clarify if you used this analysing method or how you judged that the sample size was enough for your study.
--	--

	Page 8 line 43. Was the pilot interview included in the study? Did you make any changes in the interview guide? Page 9 The four steps of Malteruds analysing process is well described from the reference. However, Figure 1 could include examples of how you did this by including text from your data. Ethics line 38, remove “ed” in consent: All participants gave informed consent. Results Page 10 line 24 “This indicates insufficiency...” would be better placed in the discussion. You have the subgroup Ignorance and turn it into the category Individual understanding and knowledge. This step is not understandable. Are there more subgroups that are not named? The same with the category Availability and knowledge in health care with the subgroup Difficult to get help. I think the analysing process needs one more turn to set the categories, before the result section is clear. Therefore, I decided not to review the discussion part. Table 2. Background factors... I don't understand why you include so many different measures, what do they add to the study? It would be more interesting to read more demographics, as working status, marital status, educational status... Also, I wonder about the self-assessed severity at 24 years. Some of the participants are 22-23 years old? Once again, the study highlights an important subject, but need to be deeper processed before publication.
--	--

REVIEWER	Howells, Laura University of Nottingham Centre for Evidence Based Dermatology
REVIEW RETURNED	15-Oct-2020

GENERAL COMMENTS	This manuscript was an enjoyable read and a welcome addition to the qualitative literature on young people's experiences of atopic dermatitis/eczema. I found the methodological quality and reporting to be of a high standard overall. The findings were interesting and insightful. The discussion was useful for contextualising the findings and understanding the clinical relevance. Whilst the standard of English is generally very good throughout the manuscript, I noticed a few small errors that I have tried to highlight within my review. Below I outline some suggestions that I hope will enable the authors a chance to clarify reporting within their manuscript: Title/focus of study:  • “transition” – the study title and study objective states that it is looking at transition. After reading your manuscript, I felt this as a central concept was perhaps a little misleading. ‘Transition’ to me typically refers to the transition in care (e.g. from child/adolescent services to adult services), and this is what I think the EAACI guidelines (reference 3 in your article) also refer to. Whilst I agree
---

that considering how your findings relate to transition between services, and may offer useful insights into how transition processes may be improved, I did not feel that the questions in your interview guide were really asking about this, and I think your focus was more on young people 'transition' to managing their AD/eczema for themselves as young people, which is different to when their parents are doing the majority of treatment and management when they are younger. I would suggest reconsidering wording in title and/or making the focus of your study very clearly about self-management. Although, it may be interesting to also reflect on what this may mean for transitioning to adult healthcare services.

Abstract (page 3):

- "persistent AD was defined as preschool.school age onset with dry skin in combination with itchy rash of typical localisation" – This description does not do justice to the measures you took to confirm AD diagnosis that are outlined more detail in the manuscript. I would consider re-phrasing.
- "TGCs" – please use full word i.e. not the abbreviation for the abstract.

Article summary (page 5):

- "Data from a longitudinal population-based birth cohort" – can you also briefly explain here what you see as key benefit/s of this?
- Wording of final bullet – consider re-phrasing. Suggestion: "The results may not be transferred to young adults with more severe atopic dermatitis requiring specialist care."

Introduction (Page 6):

- Paragraph 1 – rephrase "chronical" to "chronic"
- Paragraph 1 – You discuss discordance between patient and physician assessment of severity. Can you explain in what way they are discordant and why this is relevant to your study?
- Paragraph 2 – "topical glucocorticoids" (TGC) – my understanding is that these treatments in the UK at least are most commonly referred to as topical corticosteroids (TCS). Have you chosen TGC instead of TCS for a reason? Could you consider using TCS or maybe put the alternative name of TCS in brackets?
- Paragraph 3 – you say special transition programmes have been developed for other diseases but not AD – is this in Sweden/Europe specifically you are referring to or internationally? It would be good to be specific about this.

Methods (page 7):

- Paragraph 3 – you describe a strategic sampling procedure but use a random sampling approach rather than purposeful sampling which is commonly used in qualitative research. I can see that your approach was successful in resulting in a range of participant characteristics. Could you perhaps add a justification for why you chose this approach / perhaps provide references supporting the use of this sampling approach in qualitative studies?

Methods (page 8):

	 • Paragraph 2 – It is great to see that you have reported that the interviews were performed by an experienced nurse working in the paediatric unit. Could you provide the author initials so it is known who conducted these? Could you also add what experience/training the interviewer had in qualitative research? Can you also add something about the relationship between the interviewer and participants (i.e. was the interviewer already known to the participants? Does the interviewer deliver care to the participants/part of their team?) Methods (page 9):  • Paragraph 1 – Can you specify by initials which authors did/were involved in each stage of the data analysis. • Paragraph 1 – I found the ‘editing analysis style’ very interesting – can you provide a reference for this approach? • Did you get participant feedback on the results? If you are able to do so (i.e. have permission to contact them), I would highly recommend doing and including their feedback in your publication. Results (page 10):  • Paragraph 3 – Sub-theme title “Ignorance” needs rewording in my opinion. Whilst I think you have used this term to mean the definition of ‘lacking knowledge or information’, ignorance is used in English (UK) as a slang term as an insult to describe somebody as prejudiced/refusing to listen to others/learn. Whilst this is not your intention, I think this wording implies a value judgement about the young adults that I do not think you intend to make. I would consider rewording. Suggestion: ‘lack of knowledge’ Results (page 11):  • Paragraph 3 – Here you refer to what ‘participants have been advised’. I would be careful here with wording as you do not know exactly what participants have been advised, but you know what they report being advised (which is based on their recall of this advice.) It might also be helpful if you can indicate here which advice is in line with current best practice/guidelines and which is not?
--	---

VERSION 1 – AUTHOR RESPONSE

Reviewer: 1

Dr. Parker Magin, University of Newcastle Comments to the Author:

Important contexts for the topic of atopic dermatitis (AD) management are that it is a chronic disease that usually commences quite early in childhood and not infrequently continues into adulthood. An additional important feature is that AD involves a great deal of self-management (quite often involving ongoing regular topical treatment and skin care). Not only is this self-management onerous, it is predicated on a good understanding of the nature and natural history of the condition and on the rationale for topical treatment.

Having a child with AD is often demanding and time-consuming for parents, who need the requisite understanding of the condition and commitment to its ongoing management. The transition of this responsibility for care from parents to patients as they acquire age-related preparedness is a particularly important, as well as an interesting, period.

The current study aims to explicitly address this transition period, but the overall impression is of a fairly superficial exploration of AD in young adults rather than an exploration of transition.

We agree with the reviewer that transition is a very interesting period. However, we also agree that the present study primarily describes what it is like to live with AD as a young adult. We have therefore rephrased the title and the aim.

The study looks to have been competently performed. One issue is that the authors describe a very quantitative approach to recruitment: ‘...a strategic selection was performed.every third of these participants was invited to the interview study from a random list...’. A purposive rather than random sampling would be appropriate for a qualitative study such as this. A maximum variation sample rather than random sample would better meet the study aims.

We are sorry that the information regarding recruitment was unclear and misleading. We did perform a purposive selection, as we wanted to include participants with persistent eczema (lifelong experience, i.e., preschool/school-age onset of living with eczema). We also wanted to gather experiences from participants with different disease severities (mild, moderate, and severe/very severe eczema) in order to get a wide range of data.

We have now clarified this. The rephrased paragraph reads as follows: “A purposive selection was performed to capture young adults with experience of persistent AD (preschool/school-age onset), with various severity (mild to severe/very severe) (Table 1). Moreover, we strived to capture an equal proportion of men and women with different severity of AD. They were invited to the interview study by a clinical research nurse over the telephone. Thereafter, the first author contacted the participants, informed them about the study, and asked if they wanted to share their experiences of living with AD.” Lines 4–12, page 8.

The findings are somewhat disappointing. The Results contains a range of findings which relate to current (early adult) experience of AD. These are (as the authors discuss) consistent with previous qualitative research on AD. And the findings are general – the actual transition process is not explored beyond that the ‘participants did not state any experiences of transfer from paediatric to adult healthcare.’ And deficits in participants’ current knowledge etc were attributed (by the authors) to a failure of transition. The transition from childhood to adult AD is far more complex than the transfer of healthcare system, and current young adult experience of AD will be subject to multiple influences in addition to the transition process.

The result is more a study of AD in early adulthood rather than a study of transition in AD. The findings in themselves are interesting, if not novel.

Thank you for this very important comment. After working through the material once more, we have rephrased the aim of the study to focus on self-management instead of transition. One important finding of our study is that young adults do not have sufficient knowledge to manage and treat their AD adequately. Knowledge and understanding are of major importance to be able to perform self-management and one reason for the deficiency may be shortcomings in the transition process.

We have changed the title, as mentioned above, as well as the purpose, to focus on the concept of self-management. The latter reads as follows: “The aim of this qualitative study was therefore to explore young adults’ perceptions of living with AD in relation to the concept of self-management.” Lines 10–12, page 7.

Minor issue.

‘Cortisone’ is regularly referred to where it should be ‘corticosteroids’.

We have rephrased to “topical corticosteroid” and used the abbreviation TCS in the text on pages 12 and 13.

Reviewer: 2

Dr. Charlotte Angelhoff, Linkopings universitet Campus Norrkoping, Linkopings Universitet Institutionen for klinisk och experimentell medicin Comments to the Author:

Thank you for inviting me to review this study. This study is interesting and important. However, some major revision should be considered.

Abstract: Under results you use the abbreviation TGC. Please spell out.

Thank you for being observant. We have changed and rephrased (TGC) to (TCS) topical corticosteroid accordingly: "Their uncertainties and feelings affected how they used topical corticosteroids." Line 22–23, page 3.

Introduction

Page 6, line 38. In what way can the treatment be challenge for individuals with AD? Please, give some examples I don't think that the rational is clear enough to explain why this study is needed.

AD is an intermittent chronic disease that requires daily topical treatment (once or twice a day). This can be time-consuming, and it can also be difficult to stick to the treatment strategy. To understand the course of the disease and the treatment goals, it is important to have sufficient knowledge, which requires education and training.

We have clarified this challenge in the text and the sentence now reads as follows: "The first-line treatment of inflammation consists of topical corticosteroids (TCSs), with differing potency depending on age, localisation, and severity of the symptoms. The chronic, recurrent nature of the disease and the often- time-consuming treatment, to be performed every day, can be a challenge for individuals with AD." Lines 18–22, page 6.

We have clarified the rationale for the present study:

"To ensure self-management in AD, it is important that the individuals have sufficient skills related to treatment principles and are aware of the goals of treatment, which are to increase disease control and to extend asymptomatic periods as much as possible. This requires knowledge and education", lines 13–17, page 6, and "Overall experiences of how young adults (from the general population) with AD manage the disease are scarcely studied." Lines 8–10, page 7.

Methods:

Overall, the methods section needs major revision. In qualitative studies it is important to explain to the reader what approach and methodology the authors used. This is unclear throughout the manuscript, which gives the study low impact. It is not enough to state that it is an inductive approach. The systematic text condensation by Malterud is an analysing process based on psychological phenomenology by Giorgi.

We do agree that it is important to be clear about the approach and methodology and we apologise for being unclear.

We have rephrased and clarified both the approach and the methodology. The text now reads: "The analysis was inductive, without any theory-driven approach to the phenomenon of self-management. We were open to the data and what the material told us, from an individual to a general level. In the analysis process, we tried to bracket our preconceptions to minimize the impact of our professional and clinical experiences. Using STC, with many similarities to the editing analysis style, we developed categories derived from the empirical data." Line 3–8, page 10.

Page 7. The setting section is not clear. It is confusing how you recruited adolescents to present study, as you have described the setting of another study, BAMSE. You could write that the study is part of a greater study with a reference to BAMSE instead.

We agree with this important comment and your suggestion and have changed the text accordingly. After revision, the paragraph now reads: "The present study is a part of the longitudinal population-based cohort BAMSE^{1,2}. A purposive selection was performed to capture young adults with experience of persistent AD (preschool/school-age onset), with various severity (mild to severe/very severe) (Table 1). Moreover, we strived to capture an equal proportion of men and women with different severity of AD. They were invited to the interview study by a clinical research nurse over the telephone. Thereafter, the first author contacted the participants, informed them about the study, and asked if they wanted to share their experiences of living with AD." Lines 20–21, page 7, and lines 4–12, page 8.

Line 54, please clarify what a study nurse is

The study nurse was the nurse who examined the participants at the clinical examination at the 24-year follow-up. We have clarified this by changing to clinical research nurse. Lines 10, page 8.

Page 8 I am concerned by the analysing process and how it was performed.

Thank you for this important remark. We have worked through the analysing process once more, which gave us the opportunity to re-evaluate the results.

Line 35. Was the experienced nurse who did the interviews working with the study participants? Was the nurse skilled in interview technique? How come the authors (researchers) did not perform the interviews?

We understand that this part was unclear, and we have changed it to make things clearer. The first author (an experienced nurse) was the one who performed the interviews. She had not met the participants before the interview sessions. In our research group, we have experience of qualitative research, from both focus groups and individual interviews³⁻⁵. Before the start of the study, the first author completed a course in interview technique, covering both theory and practice.

We have included data about who performed the interviews, i.e., the first author, and her involvement regarding the participants. The sentence now reads as follows: "The interviews were performed by the first author, an experienced dermatology nurse (SL), with no connection to the study participants before the interviews." Lines 3–5, page 9.

The concept "data saturation" (**used at page 8, line 45**) is a concept in the method Grounded Theory and implies that the interview is transcribed and analysed before the next interview is performed. This is not the procedure for other analysing methods. Please, clarify if you used this analysing method or how you judged that the sample size was enough for your study.

We chose to use the concept of saturation because it is well-known and well understood when describing that the sample size is considered to be sufficient. However, as the referee points out, according to Malterud it is more correct to use the term information power.

This is an approach which considers the aim, the sample specificity, the theoretical background, the quality of dialogue and the strategy for analyses. Taken together, these determine whether sufficient information power has been obtained. This was taken into consideration when we judged that our sample size was sufficient.

We have now rephrased the text: "Data collection continued until information power was judged to have been achieved. This is an approach which considers the aim, the sample specificity, the

theoretical background, the quality of dialogue and the strategy for analyses. Taken together, these determine whether sufficient information power has been obtained.” Lines 12–15, page 9.

Page 8 line 43. Was the pilot interview included in the study? Did you make any changes in the interview guide?

The pilot interview was not included in the analysis. Minor changes were made to the interview guide after the pilot interview. We have clarified this, and the sentence now reads as follows: “To test the interview guide, one pilot interview was conducted (not included). Minor changes were then made to the interview guide.” Lines 8–9, page 9.

Page 9

The four steps of Malteruds analysing process is well described from the reference. However, **Figure 1** could include examples of how you did this by including text from your data.

Thank you for this remark. Following our re-evaluation of the material, Figures 1 and 2 have been replaced by a table (Table 3).

Step 1. Preliminary themes	Step 2. Codes	Step 3. Subgroups	Step 4. Categories
Reminds you	Always on my mind Almost never symptom-free As long as it is not visible	Strategies Acceptance	A part of life
The care is vague	They do not have time Everyone says different things Lack of information The treatment does not work	Limited availability Limited support Uncertainty Need for information	Difficult to get help Limited knowledge
Physical effects	Not good for my body Being careful Burning and stinging	Unnatural It hurts	Impact on my body

Ethics line 38, remove “ed” in consent: All participants gave informed consent.
Thank you for being observant. This has been corrected. Line 12, page 10.

Results

Page 10 line 24 “This indicates insufficiency...” would be better placed in the discussion.

We agree and have rephrased the text and relocated it to the discussion. The sentence now reads as follows: "Altogether, this indicates an insufficiency regarding self-management, including knowledge and skills in treatment principles for the disease, which may affect young adults' possibilities to take full control of their AD." Lines 15-18, page 17.

You have the subgroup Ignorance and turn it into the category Individual understanding and knowledge. This step is not understandable. Are there more subgroups that are not named? The same with the category Availability and knowledge in health care with the subgroup Difficult to get help. I think the analysing process needs one more turn to set the categories, before the result section is clear. Therefore, I decided not to review the discussion part.

Thank you, we have worked through the subgroups and categories once more. This has helped us to re-evaluate our subgroups and categories so that they are more distinct.

The category Individual understanding and knowledge has been rephrased to "Limited knowledge" and consists of two subgroups: "Uncertainty" and "Need for information". The category Availability and knowledge in healthcare has been rephrased to "Difficult to get help" and consists of two subgroups: "Limited availability" and "Limited support". We have also re-evaluated the category Get used to it and rephrased it to "A part of life", now including two subgroups: "Strategies" and "Acceptance".

We have rephrased this in the results and replaced Figures 1 and 2 with Table 3.

Table 2. Background factors... I don't understand why you include so many different measures, what do they add to the study? It would be more interesting to read more demographics, as working status, marital status, educational status... Also, I wonder about the self-assessed severity at 24 years. Some of the participants are 22-23 years old?

Our aim with this study was to gather data from young adults with experience of both persistent AD and different severities of AD, to capture as rich stories as possible. Therefore, it was important to include how we defined persistent AD and how the participants self-assessed the severity of their disease at different follow-ups, since AD usually varies over time.

It is correct that not all participants were 24 years old at the time of the 24-year follow-up, even though this was the intention. This can be confusing, so we have rephrased and relocated the sentence, and included the mean age of the participants in the data collection. The sentence now reads as follows: "In total, 15 young adults, mean age 23,4 y were included (Table 1), which gave us rich data covering the purpose of the study." Lines 15-17, page 9.

Thank you for inviting me to review this study. This study is interesting and important. However, some major revision should be considered.

Reviewer: 3

Miss Laura Howells, University of Nottingham Centre for Evidence Based Dermatology Comments to the Author:

This manuscript was an enjoyable read and a welcome addition to the qualitative literature on young people's experiences of atopic dermatitis/eczema.

I found the methodological quality and reporting to be of a high standard overall. The findings were interesting and insightful. The discussion was useful for contextualising the findings and understanding the clinical relevance.

Whilst the standard of English is generally very good throughout the manuscript, I noticed a few small errors that I have tried to highlight within my review.

Below I outline some suggestions that I hope will enable the authors a chance to clarify reporting within their manuscript:

Title/focus of study:

“transition” – the study title and study objective states that it is looking at transition. After reading your manuscript, I felt this as a central concept was perhaps a little misleading. ‘Transition’ to me typically refers to the transition in care (e.g. from child/adolescent services to adult services), and this is what I think the EAACI guidelines (reference 3 in your article) also refer to. Whilst I agree that considering how your findings relate to transition between services and may offer useful insights into how transition processes may be improved, I did not feel that the questions in your interview guide were really asking about this, and I think your focus was more on young people ‘transition’ to managing their AD/eczema for themselves as young people, which is different to when their parents are doing the majority of treatment and management when they are younger. I would suggest reconsidering wording in title and/or making the focus of your study very clearly about self-management. Although, it may be interesting to also reflect on what this may mean for transitioning to adult healthcare services.

We have taken your advice in consideration and agree with you and your opinion about self-management. In this study, it became clear that young adults do not have sufficient knowledge to manage their treatment and self-management adequately, which may be due to shortcomings in the transition process.

We have changed the title and purpose of the study, using the concept self-management, and the aim now reads as follows: “The aim of this qualitative study was therefore to explore young adults’ perceptions of living with AD in relation to the concept of self-management.” Lines 10–12, page 7.

Abstract (page 3):

“persistent AD was defined as preschool/school-age onset with dry skin in combination with itchy rash of typical localisation” – This description does not do justice to the measures you took to confirm AD diagnosis that are outlined more detail in the manuscript. I would consider re-phrasing.

Thank you for being observant. We have rephrased the text to include details from the manuscript; the sentence now reads as follows: “Persistent AD (preschool/school-age onset) was defined as dry skin in combination with itchy rash of typical localisation in the 12 months preceding the 16-year and the 24-year follow-ups.” Lines 15–18, page 3.

“TGCs” – please use full word i.e. not the abbreviation for the abstract.

Thank you for being observant. We have changed and rephrased (TGS) to (TCS) topical corticosteroids accordingly. The sentence now reads as follows: “Their uncertainties and feelings affected how they used topical corticosteroids.” Line 22–23, page 3.

Article summary (page 5):

Data from a longitudinal population-based birth cohort” – can you also briefly explain here what you see as key benefit/s of this?

One of the key benefits of longitudinal studies is the ability to follow young individuals, with detailed information on the different phenotypes and severity of their AD. In this study, we used onset and follow-up data and self-rated severity of AD to describe and define the participants’ AD, which can vary over time and may affect how they perceive their disease. We have added this to the summary and replaced bullet number 2 with this information. The sentence now reads as follows: “Data from a longitudinal population-based birth cohort, with the possibility of obtaining detailed information about different phenotypes and severity of AD over time.” Lines 3–5, page 5.

Wording of final bullet – consider re-phrasing. Suggestion: “The results may not be transferred to young adults with more severe atopic dermatitis requiring specialist care.”

Thank you for your suggestion. We have rephrased the final bullet and the sentence reads as follows: “The results may not be transferred to young adults with more severe atopic dermatitis requiring specialist care.” Lines 11–12, page 5.

Introduction (Page 6):

Paragraph 1 – rephrase “chronical” to “chronic”. *Done, line 7, page 6.*

Paragraph 1 – You discuss discordance between patient and physician assessment of severity. Can you explain in what way they are discordant and why this is relevant to your study?

It is of importance for health-care providers to be aware of and understand what is important for individuals with AD. If we have different opinions about severity, treatment, and goals, it will be difficult to reach consensus about self-management of AD. These differences have been clarified, and the sentence now reads as follows: “However, there seems to be a discordance between patients’ and physicians’ assessments of severity, judged using different instruments. The patients reported AD as more severe when their quality of life was impaired, while the physicians reported AD as more severe if sleep was reported to be disturbed.” Lines 10–13, page 6.

Paragraph 2 – “topical glucocorticoids” (TGC) – my understanding is that these treatments in the UK at least are most commonly referred to as topical corticosteroids (TCS). Have you chosen TGC instead of TCS for a reason? Could you consider using TCS or maybe put the alternative name of TCS in brackets?

We have change TGC to topical corticosteroids (TCSs) throughout the manuscript.

Paragraph 3 – you say special transition programmes have been developed for other diseases but not AD – is this in Sweden/Europe specifically you are referring to or internationally? It would be good to be specific about this.

We have now specified the context of the transition models. The new sentence reads as follows: “In Sweden, special transition models have been developed for chronic diseases such as congenital heart diseases and diabetes (still under evaluation) but – to the best of our knowledge – not for young persons with AD.” Lines 3–5, page 7.

Methods (page 7)

Paragraph 3 – you describe a strategic sampling procedure but use a random sampling approach rather than purposeful sampling which is commonly used in qualitative research. I can see that your approach was successful in resulting in a range of participant characteristics. Could you perhaps add a justification for why you chose this approach / perhaps provide references supporting the use of this sampling approach in qualitative studies?

We are sorry that the information regarding recruitment was unclear and misleading. We did perform a purposive selection, as we wanted to include participants with persistent eczema (preschool/school-age experience of living with eczema). We also wanted to gather experiences from participants with different disease severities (mild, moderate, and severe eczema) and thereby get a wide range of data.

In order to clarify this, the paragraph has been rephrased as follows: “A purposive selection was performed to capture young adults with experience of persistent AD (preschool/school-age onset), with various symptoms (mild to severe/very severe) (Table 1). Moreover, we strived to capture an equal proportion of men and women with different severity of AD. They were invited to the interview study by a clinical research nurse over the telephone. Thereafter, the first author contacted the

participants, informed them about the study, and asked if they wanted to share their experiences of living with AD.” Lines 4–12, page 8.

Methods (page 8):

Paragraph 2 – It is great to see that you have reported that the interviews were performed by an experienced nurse working in the paediatric unit. Could you provide the author initials, so it is known who conducted these? Could you also add what experience/training the interviewer had in qualitative research? Can you also add something about the relationship between the interviewer and participants (i.e. was the interviewer already known to the participants? Does the interviewer deliver care to the participants/part of their team?)

In our research group, we have experiences of qualitative research and interview technique, from both focus groups and individual interviews³⁻⁵. Before the start of the study, the first author completed a course in interview technique, covering both theory and practice. Data about who performed the interviews, i.e., the first author, and her involvement regarding the participants has now been added to the text and the text reads as follows: “The interviews were performed by the first author, an experienced dermatology nurse (SL), with no connection to the study participants before the interviews.” Lines 3–5, page 9.

Methods (page 9):

Paragraph 1 – Can you specify by initials which authors did/were involved in each stage of the data analysis.

We have now specified the authors in the different steps of the analysis: “The text was analysed by three researchers (SL, MJ & IK) in collaboration, using systematic text condensation (STC), a thematic cross-case strategy in four steps: 1st: SL, MJ & IK, 2nd: SL & MJ, 3rd: SL, MJ & IK, and 4th: SL, MJ & IK.” Lines 20–25, page 9 and lines 1–3, page 10.

Paragraph 1 – I found the ‘editing analysis style’ very interesting – can you provide a reference for this approach?

We have included the reference for this approach and reworded the sentence, which now reads: “Using STC, with many similarities to the editing analysis style⁶, we developed categories derived from the empirical data.” Lines 7–8, page 10.

Did you get participant feedback on the results? If you are able to do so (i.e. have permission to contact them), I would highly recommend doing and including their feedback in your publication.

No, we have not received feedback from the participants on the final results. However, during the interviews, participants were asked if their answers were interpreted correctly, so the interviewer received confirmation from the respondents in that step. This have been clarified in the text: “During the interview, the participants were asked if their answers were interpreted correctly. Thus, the interviewer received confirmation from the respondents.” Lines 9–11, page 9.

Results (page 10):

Paragraph 3 – Sub-theme title “Ignorance” needs rewording in my opinion. Whilst I think you have used this term to mean the definition of ‘lacking knowledge or information’, ignorance is used in English (UK) as a slang term as an insult to describe somebody as prejudiced/refusing to listen to others/learn. Whilst this is not your intention, I think this wording implies a value judgement about the young adults that I do not think you intend to make. I would consider rewording. Suggestion: ‘lack of knowledge’

Thank you for this extremely important comment. It was not our intention to use a misleading slang term. After we have worked through the subgroups and categories once more, this subgroup has been re-evaluated to “Uncertainty” in the results, line 11, page 11, and in Table 3.

Results (page 11):

Paragraph 3 – Here you refer to what ‘participants have been advised’. I would be careful here with wording as you do not know exactly what participants have been advised, but you know what they report being advised (which is based on their recall of this advice.) It might also be helpful if you can indicate here which advice is in line with current best practice/guidelines and which is not?

We agree and have rephrased the text: “They also reported that they were advised to use TCSs sparingly, at most twice a week, without regard to the severity of the AD.” Lines 8–10, page 12. Your suggestion of referring to the treatment guidelines is relevant, and therefore, we included a sentence about this in the discussion: “To have the possibility to treat AD adequately, the individuals need to have access to the appropriate potency and amount of TCSs and to treat the AD daily until the symptoms clear. Thereafter, the TCS therapy should be changed into proactive treatment, twice a week.” Line 20–23, page 18.

References

1. Wickman M, Kull I, Pershagen G, Nordvall SL. The BAMSE project: presentation of a prospective longitudinal birth cohort study. *Pediatr Allergy Immunol.* 2002;13 Suppl 15:11-3.
2. Ödling M, Andersson N, Hallberg J, et al. A Gap Between Asthma Guidelines and Management for Adolescents and Young Adults. *The journal of allergy and clinical immunology In practice.* Oct 2020;8(9):3056-3065.e2. doi:10.1016/j.jaip.2020.05.034
3. Jonsson M, Schuster M, Protudjer JLP, Bergstrom A, Egmar AC, Kull I. Experiences of Daily Life Among Adolescents With Asthma - A Struggle With Ambivalence. *Journal of pediatric nursing.* Jul - Aug 2017;35:23-29. doi:10.1016/j.pedn.2017.02.005
4. Lagercrantz B, Persson Å, Jonsson M, Kull I. Living with a Severe Allergy: Lived Perspectives from Swedish Adolescents and their Parents. *Journal of pediatric nursing.* Jan-Feb 2020;50:e107-e112. doi:10.1016/j.pedn.2019.05.018
5. Ödling M, Jonsson M, Janson C, Melén E, Bergström A, Kull I. Lost in the transition from pediatric to adult healthcare? Experiences of young adults with severe asthma. *The Journal of asthma : official journal of the Association for the Care of Asthma.* Oct 2020;57(10):1119-1127. doi:10.1080/02770903.2019.1640726
6. Crabtree BF, Miller WL. *Doing qualitative research.* SAGE; 1999.

VERSION 2 – REVIEW

REVIEWER	Magin, Parker University of Newcastle, Discipline of General Practice
REVIEW RETURNED	14-Mar-2021

GENERAL COMMENTS	The authors have addressed the issues raised in my review. I have a concern, however, with the post hoc change in the study aim. The study may have had limited findings related directly to the original stated aim, and other findings may have been of more import, but the aim was still the aim.
---

REVIEWER	Howells, Laura University of Nottingham Centre for Evidence Based Dermatology
REVIEW RETURNED	07-Apr-2021

GENERAL COMMENTS	Thank you for your hard work addressing each of my peer review comments. I found your responses to adequately cover the concerns/queries that I had, and I think this paper now makes an interesting addition to the qualitative literature on young people living with atopic eczema.
--

VERSION 2 – AUTHOR RESPONSE

Reviewer: 1

Dr. Parker Magin, University of Newcastle Comments to the Author:

The authors have addressed the issues raised in my review.

I have a concern, however, with the post hoc change in the study aim. The study may have had limited findings related directly to the original stated aim, and other findings may have been of more import, but the aim was still the aim.

Thank you for this important comment. We agree that the overall purpose was to explore young adults' perception of living with atopic dermatitis in relation to the transition process. However, after the constructive comments we received, we reviewed the analysis once again, and it became clear to us that the result more describes young people's ability to manage their illness themselves whereas transition may be one part. Therefore, we reformulated the purpose to be more specific and hopefully clarify the content of the study to the reader.

Reviewer: 3

Dr. Laura Howells, University of Nottingham Centre for Evidence Based Dermatology

Comments to the Author:

Thank you for your hard work addressing each of my peer review comments. I found your responses to adequately cover the concerns/queries that I had, and I think this paper now makes an interesting addition to the qualitative literature on young people living with atopic eczema.

Once again, thank you for constructive comments that helped us improve the manuscript substantially.